# Sexual excitation induces courtship ultrasonic vocalizations and cataplexy-like behavior in orexin neuron-ablated male mice

Tomoyuki Kuwaki [1]✉ & Kouta Kanno [2]

Cataplexy is triggered by laughter in humans and palatable food in mice. To further evaluate mice's cataplexy, we examined courtship behavior in orexin neuron-ablated mice (ORX-AB), one of the animal models of narcolepsy/cataplexy. Wild-type female mice were placed into the home cage of male ORX-AB and cataplexy-like behavior was observed along with ultrasonic vocalizations (USVs), also known as the "love song". ORX-AB with a female encounter showed cataplexy-like behavior both during the dark and light periods, whereas ORX-AB with chocolate predominantly showed it during the dark period. During the light period observation, more than 85% of cataplexy-like bouts were preceded by USVs. A strong positive correlation was observed between the number of USVs and cataplexy-like bouts. Cataplexy-like behavior in narcoleptic mice is a good behavioral measure to study the brain mechanisms behind positive emotion because they can be induced by different kinds of positive stimuli, including chocolate and female courtship.

[1] Department of Physiology, Graduate School of Medical and Dental Sciences, Kagoshima University, Kagoshima, Japan. [2] Laboratory of Neuroscience, Course of Psychology, Department of Humanities, Faculty of Law, Economics and the Humanities, Kagoshima University, Kagoshima, Japan. ✉email: kuwaki@m3.kufm.kagoshima-u.ac.jp

Happiness and positive emotion are key for both mental and physical well-being. Although their benefits are well known, the underlying brain mechanisms are not well understood. Brain imaging studies in humans[1] predicted the possible involvement of several brain structures in happiness, such as the nucleus accumbens (NAc), ventral pallidum, and anterior cingulate cortex. However, more detailed information on the neurotransmitters and neuronal pathways involved in the interaction between emotions and bodily health is currently unavailable. In experimental animals, it is necessary to estimate their emotional state through behavioral observation because verbal communication is not possible. Previous studies have shown successful estimation of positive emotion in mice through counting the number of tongue protrusions in response to the presentation of honey[2] and measurement of facial expressions in head-fixed mice with the assistance of artificial intelligence[3].

Cataplexy is a form of muscle atonia seen in both human narcolepsy patients and animal narcolepsy models. It is thought to be a good behavioral measure of positive emotion because it is triggered by laughter in humans [4], playing in dogs[5], and palatable food in mice[6]. Cataplexy is unique in that head fixation is not necessary to observe associated behaviors and possible triggers are not restricted to food. Apart from the positive emotional context, there are many studies pertaining to cataplexy as a symptom of narcolepsy[6–14]. Deficiency of the neuropeptide orexin in the hypothalamus is one of the causes of this symptom[7] and activation of the NAc and the prefrontal cortex is thought to be an upstream trigger[9,10]. Studies have also shown that the amygdala is a key brain structure involved in the occurrence of cataplexy[6,11,12]. Cataplexy is mainly observed during the dark period, the active period in nocturnal mice, but the reason is still unknown[7,9,13,14].

In order to expand on these findings, we decided to characterize the positive emotion-related nature of cataplexy through experiments utilizing orexin neuron-ablated (ORX-AB) mice, an animal model of narcolepsy/cataplexy[14]. We specifically hypothesized that the introduction of a female mouse into a male ORX-AB mouse's home cage would induce positive excitation and therefore induce cataplexy. Sexual excitation in response to a female mouse can be quantified by measuring courtship ultrasound vocalizations (USVs), one of the precopulatory behaviors in male mice[15,16] and thought of as a "love song"[17]. These courtship USVs are considered an index of sexual motivation independent of male copulatory performance[18] and are enhanced by sociosexual experience[19]. In addition, replacement of testosterone into the ventral tegmental area of castrated male mice, one of the main dopaminergic sources, restored the number of courtship vocalizations back to precastration levels[20]. These findings suggest that male to female USVs can be a measure of positive excitation.

During the course of this study, Hung et al.[21] reported that an animal model lacking both orexin neurons and melanin concentrating hormone (MCH) neurons lose circadian regulation of cataplexy and thus show high incidence of cataplexy during both the dark and light period. Since we found a high incidence of cataplexy in ORX-AB mice during the light period as a result of a female encounter (see Fig. 1 and related "Results" section), we also examined whether MCH neurons were preserved by utilizing immunohistochemistry.

## Results

### Time-dependent occurrence of cataplexy-like behavior. In mice narcolepsy models, most cataplexy occurs during the nighttime[7,9,13,14]. However, any possible effects on the frequency of cataplexy resulting from encountering a female during the

daytime have not been reported. To explore this, we examined the frequency of cataplexy-like behavior during the dark and light periods. Either a female mouse or a piece of chocolate was introduced to the home cage of male ORX-AB mice at the beginning of either the dark period or the light period (Fig. 1a).

During the dark period with regular chow, ORX-AB mice showed ~5 bouts of cataplexy-like behavior every 2 h (Fig. 1b) and $31.7 \pm 2.6$ total bouts ($n = 6$) over 12 h. During the light period, the same ORX-AB mice with regular chow showed almost no bouts of cataplexy-like behavior in any time bin (Fig. 1c) and the total number of cataplexy-like bouts over 12 h was only $0.7 \pm 0.2$. Sidak's multiple comparisons test revealed a significant difference between female introduction and regular chow at 0–2, 2–4, and 4–6 ZT. Food consumption during the light period ($1.6 \pm 0.2$ g) was significantly less than during the dark period ($3.4 \pm 0.6$ g, $P = 0.014$, paired $t$-test).

When the data for 12 h (the entire day or night) were treated as a whole (Fig. 1d), two-way ANOVA (treatment: regular chow, chocolate, female × period: dark, light) revealed that there were significant differences in the number of cataplexy-like bouts among treatments ($F_{2,15} = 13.22$, $P = 0.0005$) and between periods ($F_{1,15} = 99.19$, $P < 0.0001$). The interaction was not significant ($F_{2,15} = 1.893$, $P = 0.1850$). Chocolate and female encounter both significantly increased the number of cataplexy-like bouts compared to regular chow. However, the effect of the female encounter was short-lived because the difference between female encounter and regular chow was significant only in the first half of the observation period (Fig. 1c). The consumption of chocolate during the light period ($1.3 \pm 0.1$ g) was significantly lower than during the dark period ($2.5 \pm 0.1$ g, $P < 0.0001$, paired $t$-test), whereas the consumption of regular chow was not different ($0.4 \pm 0.1$ g during light period and $0.3 \pm 0.0$ g during dark period, $P = 0.092$).

Thus, we confirmed that more frequent bouts of cataplexy-like behavior can be observed during the dark period irrespective of which type of stimulation the mice are receiving (chocolate or female). In addition, we found that introduction of a female can increase frequency of cataplexy-like behavior during the light period when chocolate did not show a cataplexy-increasing effect.

### Close relationship between cataplexy-like behavior and USVs during the light period. To examine whether the increase in cataplexy-like bouts resulting from a female encounter was caused by its positive emotion-inducing effect, we recorded USVs which are putative "love songs" that occur during courtship[17], along with video of cataplexy behavior (Fig. 2a). Using information from the previous result, we decided to record USVs and cataplexy behavior for 4 h from 11:00 a.m. (ZT4) to 3:00 p.m. (ZT8), a period when cataplexy bouts were expected to be the least frequent in the absence of any stimulation.

During the 4 h observation period, the number of USV calls measured during sessions when ORX-AB mice were exposed to a female ($3688 \pm 556$ calls) was significantly greater than those to a male encounter ($1323 \pm 767$ calls) (Fig. 2b). While the average frequency of USV calls was not different between the two (Fig. 2c), the average duration of USV calls was significantly longer with female than with male encounter (Fig. 2d).

Similar to the number of USV calls, ORX-AB mice showed more cataplexy-like bouts (see supplemental videos for example) with a female ($14.7 \pm 3.6$ bouts) than with a male ($2.7 \pm 1.3$ bouts, $P = 0.037$, $n = 6$, paired $t$-test). Correlation analysis between the number of USV calls and the number of cataplexy-like bouts observed in each recording revealed a significant ($P = 0.0025$) positive correlation ($r^2 = 0.6166$) between the two (Fig. 3a).

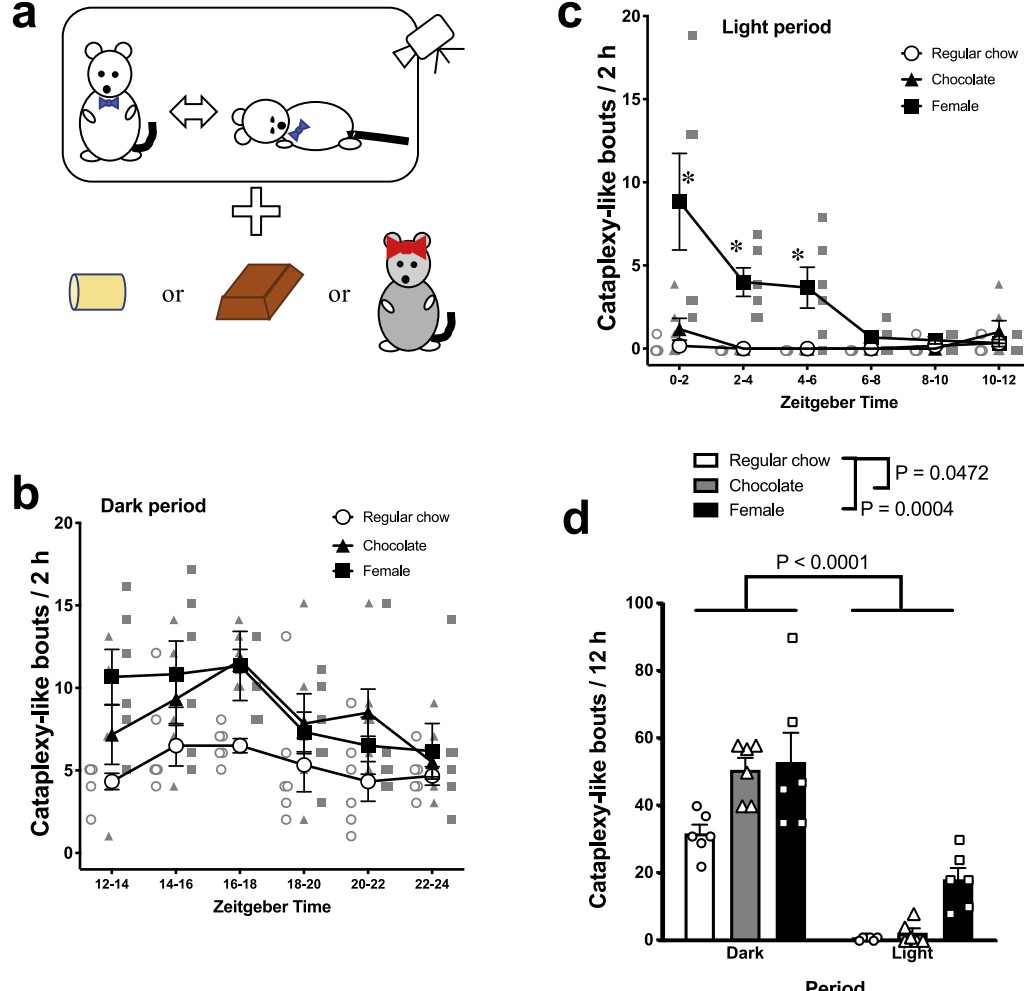

**Fig. 1 Time-dependent occurrence of cataplexy-like behavior and the effect of chocolate and female encounter. a** Schematic explanation of the experiment. Behavior of orexin neuron-ablated male mouse was video observed with regular chow, chocolate, or female for 12 h. **b** The number of cataplexy-like bouts during the dark period (ZT12–ZT24) in three treatment groups. Two-way ANOVA (treatment: regular chow, chocolate, female × period: bin length of 2 h) revealed that there was a significant difference in the number of cataplexy-like bouts during the dark period among treatments ($F_{2,15} = 4.094$, $P = 0.0381$) and between time points ($F_{5,75} = 5.723$, $P = 0.0002$). The interaction was not significant ($F_{10,75} = 1.148$, $P = 0.3395$). **c** The number of cataplexy-like bouts during the light period (ZT0–ZT12) in three treatment groups. There was a significant difference among treatments ($F_{2,15} = 20.85$, $P < 0.0001$) and between time points ($F_{5,75} = 6.639$, $P < 0.0001$). The interaction was also significant ($F_{10,75} = 5.368$, $P < 0.0001$). **d** Summary for light/dark period over 12 h. Overall, the number of cataplexy-like bouts was higher during the dark period than during the light period ($F_{1,15} = 99.19$, $P < 0.0001$). Note that both chocolate and female encounter increased the number of cataplexy-like bouts during the dark period compared to regular chow. Only female encounter increased the number of cataplexy-like bouts during the light period. The effect of the female encounter on number of cataplexy-like bouts was short-lived and gradually decreased toward regular chow level over 6 h. Data are expressed as mean ± SEM. $n = 6$ for each group. *$P < 0.05$ as compared to regular chow.

To examine a more detailed relationship between the two parameters, we have divided the cataplexy-like behavior into two categories: (1) preceded by USVs within 1 min before onset and (2) started without USVs. With a female encounter, most cataplexy-like bouts were associated with preceding USVs (85%, 75/88 bouts), whereas with male encounter, the ratio was significantly smaller (44%, 7/16 bouts, $P = 0.0008$, Fisher's exact chi-square test). When we reanalyzed the correlation between the number of USVs during the 1 min preceding cataplexy-like behavior with a female encounter and the number of USV-associated cataplexy-like bouts, we found a much stricter correlation between the two parameters ($r^2 = 0.9606$, $P = 0.0006$) (Fig. 3b).

Two-way ANOVA revealed that there was a significant difference in the number of cataplexy-like bouts between female

and male encounters ($F_{1,20} = 7.961$, $P = 0.0105$, Fig. 3c), the single cataplexy-like episode duration ($F_{1,20} = 5.947$, $P = 0.0242$, Fig. 3d), and the total time spent in cataplexy-like behavior ($F_{1,20} = 6.085$, $P = 0.0228$, Fig. 3e). Interaction (male/female × with/without preceding USV) was significant for the number of cataplexy-like bouts ($F_{1,20} = 6.290$, $P = 0.0209$) and total time spent in cataplexy-like behavior ($F_{1,20} = 4.428$, $P = 0.0482$), but not for single cataplexy-like episode duration ($F_{1,20} = 2.050$, $P = 0.1676$). With female encounters, the number of cataplexy-like bouts was significantly higher in it with USV than that without USV (Fig. 3c), but there was no statistically significant difference in single cataplexy episode duration (Fig. 3d). Therefore, the increase in total time spent in cataplexy-like behavior seemed to be mainly caused by the increase in the number of cataplexy-like

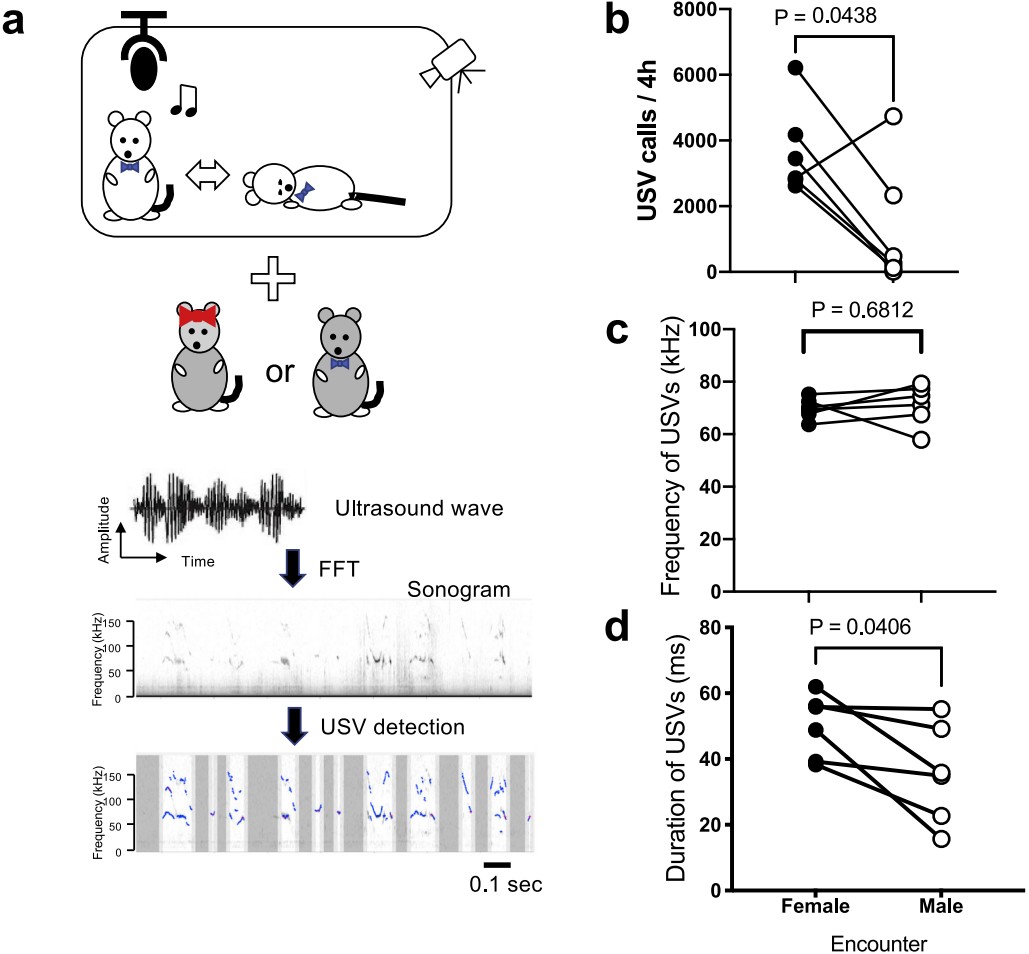

**Fig. 2 Ultrasound vocalizations in orexin neuron-ablated mice. a** Schematic explanation of the experiment. Behavior of an ORX-AB male mouse was videotaped together with recordings of the ultrasound vocalization with a female or a male encounter in his home cage for 4 h. Recorded ultrasound waves were analyzed offline to detect syllables (USV calls) indicate blue line in the bottom figure. **b** The number of USV calls. **c** Average frequency of USV calls. **d** Average duration of USVs. Each ORX-AB mouse ($n = 6$) was recorded twice; one with a female encounter and another with a male encounter. $P$ values in the figure were obtained by paired $t$-test.

bouts. In sum, there seems to be a close relationship between occurrences of USVs and occurrences of cataplexy-like behavior when we look at it in a time range of hours (Fig. 3a) and in individual episodes (Fig. 3b–e).

We also analyzed the number of USVs by separating and counting the USVs that occurred preceding cataplexy-like behavior and those that occurred without cataplexy-like behavior. With a female encounter, USVs associated with cataplexy-like behavior occurred at $10.8 \pm 1.8$ calls/min and those independent from cataplexy-like behavior occurred at $14.7 \pm 2.1$ calls/min. There was no significant difference between the two categories of USVs ($P = 0.215$, paired $t$-test). Therefore, the preceding USVs do not seem to be a special type of vocalization that are only associated with cataplexy-like behavior.

**Ablated orexin neurons and conserved MCH neurons in ORX-AB mice.** To confirm ablation of orexin neurons and conservation of MCH, we examined the brains of wild-type mice (Fig. 4a) and ORX-AB (Fig. 4b) after the completion of behavioral testing and USV measurements. As expected, no orexin immunoreactivity was observed in ORX-AB mice, whereas wild-type mice showed $916 \pm 11$ ($n = 6$) positive cells. The number of MCH-positive cells was not different between ORX-AB mice ($1123 \pm 16$,

$n = 6$) and wild-type mice ($1136 \pm 29$, $P = 0.705$), indicating specific ablation of orexin neurons in ORX-AB mice.

### Discussion

**Occurrence of cataplexy-like behavior was time-dependent under regular chow and chocolate conditions, but female encounter overrode the time dependency.** In this study, we found that occurrence of cataplexy-like behavior in ORX-AB mice was regulated by circadian rhythmicity because cataplexy mainly occurs during the dark period even with chocolate (Fig. 1). Although the consumption of chocolate during the light period ($1.3 \pm 0.1$ g) was about half of that during the dark period ($2.5 \pm 0.1$ g), this seemed to not fully explain the observed decrease in the number of cataplexy-like bouts during the light period (less than 5% of the dark period; $2.2 \pm 1.3$ vs. $50.0 \pm 3.8$ bouts). Alternatively, the introduction of a female mouse into the male's home cage increased the frequency of cataplexy-like behavior in the resident male ORX-AB mouse even during the light period when ORX-AB mice seldom showed cataplexy-like behavior even with palatable food. Thus, the power of the female encounter to induce cataplexy-like behavior seemed much stronger than the ingestion of chocolate. Although the cataplexy-inducing effect of a female encounter gradually decreased over 6 h, presumably

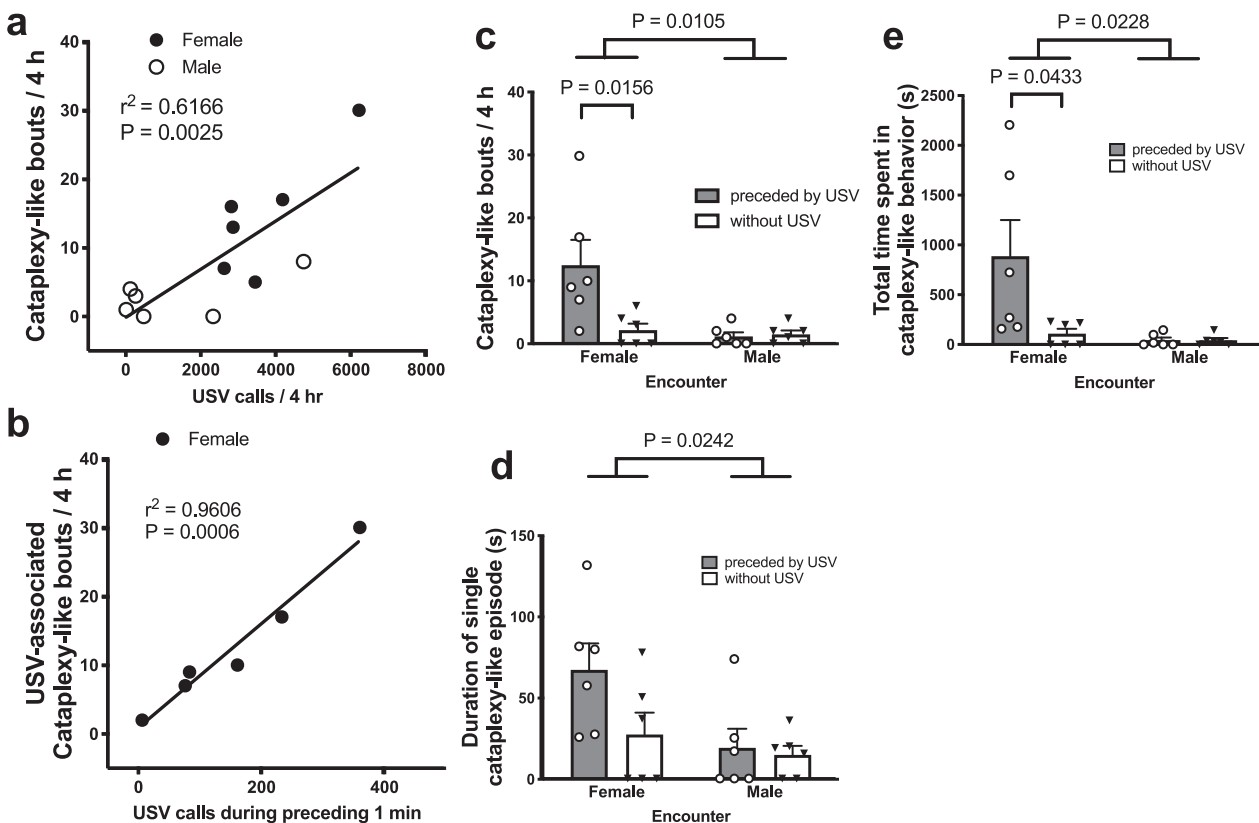

**Fig. 3 Relationship between cataplexy and ultrasound vocalizations. a** Correlation analysis between the number of ultrasound vocalization calls (USV) and the number of cataplexy-like bouts showed a positive correlation between the two. **b** Correlation analysis between the number of calls during the 1 min preceding cataplexy and the number of USV-associated cataplexy-like behavior. A much stronger correlation was observed than in **a**. Characterization of the number (**c**), duration (**d**), and total time (**e**) of cataplexy-like bouts. These parameters are classified according to the sex of the mouse encountered and whether it is preceded by USVs within 1 min before the onset of cataplexy-like behavior. Data are expressed as mean ± SEM of six ORX-AB mice. Each animal was tested twice (with female and male). *P* values in the figure were obtained using two-way ANOVA and following Sidak's multiple comparison test.

from acclimatization, its initial impact during the first 2 h was not different between the light and dark periods (8.8 ± 2.9 bouts during the light and 10.7 ± 1.7 bouts during the dark period). The overriding effect the female encounter had on the light/dark cycle was not caused by a deficiency of MCH neurons in the ORX-AB mice because the number of MCH-positive neurons was comparable to that of wild-type mice (Fig. 4).

**Characteristics of USV in the ORX-AB mice.** Previous reports using WT male mice showed 100-300 USVs/min with female encounter[22–25]. In the present study, we observed 3000–4000 USVs/4 h (Fig. 2b), which roughly corresponds ~15 USVs/min. However, USV calls (and incidence of cataplexy-like behavior, see Fig. 1c) decreased as time passed and a mouse probably acclimatized to the encounter. When we focused on the first 3 min in the recording period (many studies on courtship USV record up to several min), USVs recorded in ORX-AB male–WT female situation was 108.3 ± 27.5/min (*n* = 6). Thus, ORX-AB male mice seemed sensitive to positive rewards within the range of WT mice, at least in the current study paradigm. Observed frequency in both resident male–intruder female and resident male–intruder male USVs was comparable (Fig. 2c) and similar to the previously reported values in several genetic strains of mice[22,26]. Meanwhile, average duration of male–female USVs was significantly longer than that of male–male USVs (Fig. 2d). Long-duration calls of male–female USVs are known to be observed in latter phase of sexual behavior where mounting often occur[27]. Therefore, longer

duration in the male–female condition here was thought to reflect strong sexual excitation.

**Cataplexy-like behavior during the light period was closely related to the USVs.** We found for the first time that the occurrence of cataplexy-like behavior was closely related to that of USVs in both an overall manner by using correlation analysis (Fig. 3a) and an episodic manner by separating cataplexy-like bouts by whether or not they were preceded by USVs (Fig. 3b–e). With female encounter particularly, 85% of cataplexy-like bouts were preceded by USVs, indicating a causative relationship between USVs and cataplexy-like behavior. However, not all USVs were followed by cataplexy-like behavior and some cataplexy-like behavior occurred without preceding USVs. Thus, we propose that a shared common cause, most likely positive excitation in response to a female encounter, induced both USVs and cataplexy-like behavior in male ORX-AB mice. In regard to this finding, it is interesting to note that Burgess et al.[28] reported, although only in a meeting abstract form, that the number of USVs was positively correlated with the number of cataplexy bouts in female orexin-knockout mice. If these USVs were a result of encountering a male then we can infer that they may induce cataplexy, but it is difficult to suppose that male USVs always induce positive excitation in female mice. In addition, we can assume in their study, similar to the present study, that not all USVs were associated with cataplexy and thus other factor(s) are most likely also involved in the induction of cataplexy. Nevertheless, social interaction-associated USVs in their study would be

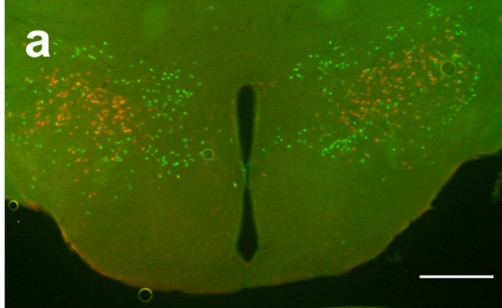

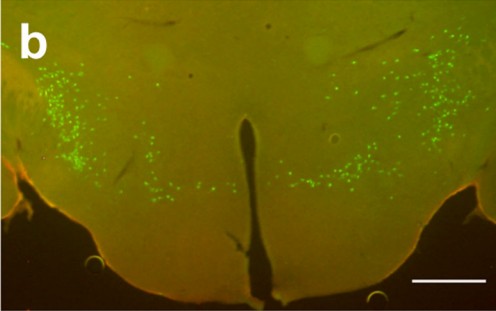

**Fig. 4 Localization of orexin and MCH-positive cells in the hypothalamus.**
**a** Typical photograph of the hypothalamus of a wild-type mouse showing the distribution of orexin-positive cells (red) and melanin concentrating hormone (MCH)-positive cells (green). **b** Typical photograph from ORX-AB mouse showing there are no longer any orexin-positive cells. Scale bar: 500 μm.

related to positive but not negative emotions. It is plausible to say that positive emotion induces USVs and cataplexy-like behavior.

These results indicate that a female encounter is a powerful trigger that induces positive excitation and cataplexy-like behavior in male ORX-AB mice. Occurrences of cataplexy-like behavior in narcoleptic mice are a good behavioral measure to study the brain mechanisms behind positive emotion because they can be observed in freely moving animals and can be induced by different kinds of positive stimuli, including chocolate and female courtship.

## Methods
**Ethics approval**. All experiments were conducted at Kagoshima University in accordance with the guiding principles for the care and use of animals in the field of physiological sciences published by the Physiological Society of Japan (2015) and were approved by the Experimental Animal Research Committee of Kagoshima University (MD17105).

**Animals**. For the animal model of narcolepsy, 24 male ORX-AB mice[10,14] that were 8–12 weeks old at the start of the experiment and weighed 24–32 g were used. The original mating pairs were a generous gift from Prof. Yamanaka at Nagoya University and were bred in Kagoshima University's animal facility. A method for selective ablation of orexin neurons has been previously reported[14]. In short, orexin-tetracycline transactivator (tTA) mice, which express tTA exclusively in orexin neurons under the control of the human prepro-orexin promoter[14], were bred with tetO diphtheria toxin A fragment (DTA) mice (B6.Cg-Tg (tetO DTA) 1Gfi/J, The Jackson Laboratory) to generate orexin-tTA; tetO DTA mice. In these double transgenic mice (called ORX-AB in this paper), doxycycline is removed from their chow starting from birth so by 4 weeks of age, almost all (>97%) of orexin neurons are ablated[14]. Ablation of orexin neurons was confirmed in our previous studies[10,29] and in the current study. Although both male and female ORX-AB mice show cataplexy, we used only male ORX-AB mice in this study because most vocalizations are produced by the male and the vocal contribution of the female is very limited in male–female courtship behaviors[23,30,31]. We used C57BL/6J mice of both sexes (8–12 weeks old) as the intruder to the resident home cage of ORX-AB mice. For both wild-type and ORX-AB mice, sexually unexperienced male and female mice were used to avoid possible modulation of sexual behaviors by experience[19,32]. All mice were housed in a room that was maintained

at 22–24 °C with lights on at 7:00 a.m. and off at 7:00 p.m. Mice had food and water available ad libitum.

**Behavioral observation of cataplexy**. On the experimental day, individually housed ORX-AB mouse in their home cage were placed into a soundproof recording box. The chamber was illuminated with a far infrared lamp (940 nm, SA2-IR, World Musen, Hong Kong) during the dark period and with a visible light lamp during the light period. Mouse behavior was continuously recorded with a video camera (CBK21AF04, Imaging Source Asia, Taipei, Taiwan) and monitored on a personal computer located outside the soundproof box using the video capture function in LabChart (ADInstruments Inc., Bella Vista, NSW, Australia). Cataplexy was determined according to the established criteria for mice[33] which is defined by several observable features. The first feature is an abrupt episode of nuchal atonia lasting at least 10 s. Atonia was determined to occur when mice were in a prone position with their head and belly down in the bedding with their limbs and tail typically situated straight out from the trunk. This posture shows a clear contrast to a normal sleeping position in which mice are curled up and fold their limbs and tail underneath their trunk. Second, the mouse is immobile aside from the movements associated with breathing during an episode. Finally, there must be at least 40 s of active wakefulness (moving) preceding the atonia episode. Original criteria recommend recordings of EEG but we did not adopt EEG to avoid possible obstruction of social interaction by the recording cable. Therefore, we call "cataplexy-like behavior" instead of "cataplexy" in this manuscript.

**Recording of ultrasonic vocalizations**. Ultrasonic sounds (see Fig. 2a for the outline of the processing) were detected using a condenser microphone (Ultra-SoundGate CM16/CMPA, Avisoft Bioacoustics, Berlin, Germany) designed for recordings between 10 and 200 kHz. The microphone was placed at a height of 20 cm from the floor and connected to an ultrahigh-speed ADDA converter BSA768AD-KUKK1710 and its software SpectoLibellus2D (Katou Acoustics Consultant Office, Kanagawa, Japan) with a sampling rate of 384 kHz (to measure 20–192 kHz). Recorded sounds were saved on a computer as wav files using SpectoLibellus2D. These sound files were analyzed with the GUI-based software USVSEG (implemented as MATLAB scripts) developed by us[34]. Continuous sound signals with frequencies ranging from 40 to 160 kHz and durations ranging from 3 to 300 ms were analyzed and detected as syllables (USV calls). Devocalizing encounters were not tried because previous studies suggested that most vocalizations are produced by the resident male of the home cage and that the vocal contribution of the female is limited in the male–female courtship behaviors[23,30,31] and the contribution of an intruder male is also limited in the male–male USVs[23].

**Immunohistochemistry**. Ablation of orexin neurons and conservation of MCH-containing neurons were examined using the method reported[29]. In brief, mice were deeply anesthetized with urethane (1.8 g/kg, i.p.) and transcardially perfused with phosphate-buffered saline (PBS, 0.01 M, pH 7.4), followed by 4% paraformaldehyde (PFA) in PBS. The brain was removed, postfixed in 4% PFA solution at 4 °C overnight, and immersed in 30% sucrose in PBS at 4 °C for 2 days. Coronal sections, including the hypothalamus, were cut at 40 μm thickness using a vibratome and every fourth section (ten slices in a mouse) was used for immunostaining. The brain sections were immersed in blocking solution (1% normal horse serum and 0.3% Triton-X in 0.01 M PBS) for 1 h at room temperature. The sections were then incubated with primary antibodies overnight. Primary antibodies were diluted in blocking solution and the conditions were as follows: rabbit anti-orexin antiserum (1:1000, 14346-v, Peptide Institute, Ibaraki-Shi, Osaka, Japan) and goat anti-MCH antiserum (1:100, sc-14507, Santa Cruz Biotechnology, Inc., Santa Cruz, CA, USA). The sections were washed with PBS and then incubated with the secondary antibodies CF568-labeled donkey anti-rabbit IgG (1:500, 20098, Biotium, Hayward, CA, USA) and CF488-labeled donkey anti-goat IgG (1:500, 20016, Biotium). The sections were then mounted on a glass slide and examined using a fluorescence microscope (BZ-X700, Keyence Corp., Osaka, Japan).

**Experimental design**. Experiment 1: Examination of dark/light period and time dependency of occurrence of cataplexy-like behavior.

To examine whether the cataplexy-increasing effects of palatable food and female encounter were time-dependent, 18 ORX-AB mice were divided into three groups (n = 6 each) and behavior related to cataplexy was observed for the 12 h of the light period and the 12 h of the dark period. Mice were individually housed for more than 2 weeks before measurement and bedding materials were not changed before completion of the experiment. At the start of recording session, milk chocolate (one Hershey's Kiss; Hershey) was given in addition to regular chow to the first group, and an intruder female was introduced to the second group. Only regular chow and water was given to the third group which served as a control. An interval of more than 3 days was set between light period and dark period recordings and the order of light/dark recording sessions was randomized (three mice experienced the light period first and separate three mice had dark period recording first) to avoid possible confounds resulting from previous experience.

Experiment 2: Relationship between ultrasonic vocalizations and cataplexy-like behavior.

Based on the findings from experiment 1, ultrasonic vocalization and cataplexy-like behavior were recorded for 4 h from 11:00 a.m. (ZT4) to 3:00 p.m. (ZT8) when cataplexy-like behavior was least likely to occur if no other mice were introduced. Male ORX-AB mice ($n = 6$) were used for this experiment and both female and male wild-type mice were used for encounters. For randomization, three ORX-AB mice were first examined with female encounters and the remaining three ORX-AB mice had male encounters. After an interval of over 3 days, the groups were again examined following an encounter with the opposite sex they initially encountered during the first examination. After completion of the experiment, ORX-AB mice and male wild-type mice used during the encounters were utilized for histological examination of orexin and MCH.

**Statistics and reproducibility**. Statistical analyses were performed using Prism software v.6 (GraphPad). A two-way ANOVA followed by Sidak's multiple comparison test was used unless stated otherwise. For correlation analyses, the Pearson correlation coefficient was calculated. Fisher's exact chi-square test and Student's $t$ test were also used where appropriate. $P < 0.05$ was considered statistically significant. All data are presented as mean ± standard error of the mean. Statistical analyses were performed duplicate by both authors.

**Reporting summary**. Further information on research design is available in the Nature Research Reporting Summary linked to this article.

## Data availability

The summary statistics are available within the article. All source data underlying the graphs and charts presented in the main figures are available in the Supplementary Data file. The data that support the findings of this study are available from the corresponding author upon reasonable request.

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

## Acknowledgements

This work was supported by JSPS KAKENHI Grants (16H05130 and 16K13112 to T.K. and 18K13371 and 19H04912 to K.K.). The authors would like to thank Prof. Akihiro Yamanaka at Nagoya University for his generous gift of original mating pairs of orexin-tTA mice. The authors would like to thank Jordan L. Pauli for English editing, Miki Sakoda for her excellent technical assistance, and all the members of the Department of Physiology for their support. The authors would also like to thank all the staff members of the Institute of Laboratory Animal Sciences at Kagoshima University for keeping the animals in good condition.

## Author contributions

T.K. designed the study. Both authors conducted the study, analyzed the data, wrote the manuscript, and approved the final version of the manuscript.

## Competing interests
The authors declare no competing interests.
