## [Peer Review File · Communications Biology]

Reviewers' comments:

Reviewer #1 (Remarks to the Author):

The authors characterized cataplexy induced by sexual excitation in a mouse model of narcolepsy. They found that sexual excitation induced cataplexy more frequently than other triggers, such as chocolate, even during the light period. Most of the cataplexy bouts induced by sexual excitation were preceded by ultrasonic vocalizations (USVs). Based on these observations, the authors proposed that cataplexy in narcoleptic mice is a good behavioral measure to study the brain mechanisms behind positive emotion. The findings are very intriguing. However, because it is known well that positive emotion efficiently induces cataplexy, it is easily imaginable that sexual excitation also induces cataplexy. Therefore, the scientific advances generated by the study are limited.

First of all, according to ref 28, the definition of murine cataplexy requires EEG measurements, which were lacking in the current study. Therefore, the behavioral arrests observed may be immature to be called cataplexy.

It is impressive that a female encounter could induce cataplexy during the daytime, in which chocolate can not. Is this a qualitative or quantitative difference? Do the authors think that the mechanisms to induce cataplexy are different between these two? Or is the level of excitation and positive emotion much higher with a female than chocolate? If the former is the case, it would be very nice to include some data showing the qualitative difference. The latter possibility suggests that cataplexy is a just measure of the degree of excitement, which may not be a so novel finding.

According to Fig. 2, a male encounter could induce cataplexy during the light period, although the frequency was much lower as compared to a female encounter. However, encountering an unknown male may induce negative but not positive emotion. Therefore, I think that this result is surprising considering the inability of chocolate to induce cataplexy during the daytime.

A few human studies suggested that narcolepsy patients show increased responses to positive rewards. In light of this information, did narcoleptic mice emit USV calls more than WT mice did upon a female encounter?

Reviewer #2 (Remarks to the Author):

This manuscript demonstrates that sexual arousal can trigger cataplexy in orexin ablated mice across the day-night cycle, in contrast to palatable food, which mostly triggers during the night. One challenge in the field is to identify the degree of emotional arousal in animals-- indeed, it is challenging to observe this at all.

This paper finds a clever way to use cataplexy as a kind of readout for emotional arousal. I was particularly impressed by the strong association between the number of vocalizations and subsequent attacks.

The paper is straightforwardly written and the figures are easy to understand. The stats are complete and proper, although I think some of the comparisons could be removed or moved to the figure legend, to make reading a little bit easier. The stats should complement the story and support the conclusion, but in a few places, the stats overwhelm the narrative.

I also appreciate why the authors included the MCH staining; it is fine, but I would hardly expect that anything would be found and I am not sure the manuscript needs it-- the rhythm of cataplexy remained intact to food, suggesting already there was no MCH loss as in the other recent paper.

Reviewer #3 (Remarks to the Author):

The authors are commended for their clear & valuable submission. The experimental designs and overall methodology are described fully & in excellent detail. The figures are well-crafted, showing individual data points where appropriate (e.g., Fig. 1C, etc.). I have only minor comments to modestly improve the overall impact of this generally very good manuscript.

1) An experimental design figure will help readers know exactly what the authors did in their studies. Moreover, in this figure, it is advised to include a USV spectrogram example, so that readers unfamiliar with the USV measure can better appreciate what exactly is being measured in the mice.

2) While the authors acknowledge in the Discussion literature supporting that resident male mice are the likely emitters of USV during recording session, this caveat should be additionally included in the Results text for full transparency. For instance, "ORX-AB mice emitted more USV calls with a female (3688 ± 556 calls)" should be minorly revised to "USVs measured during sessions when ORX-AB mice were exposed to a female were greater ($XXX \pm XXX$) compared to...".

3) Can a bit more information regarding the acoustic features of recorded USVs be provided? In Methods, it is described that the Avisoft condenser microphone can record frequencies up to 192-kHz. Were the detected USVs predominantly within a frequency range of 40-60 kHz as has been described as relating to "positive affective" USVs in rats? Or were the detected USVs typically much higher in frequency? Additional measures of the recorded USVs would be helpful, minimally including mean frequency of detected USVs (per group, per session) and duration (in ms).

Excellent work!

We would like to take this opportunity to express our sincere thanks to the reviewers. Their comments revealed weak points in our original manuscript and showed the way for improvement. Please find point-to-point responses below.

Reviewer #1 (Remarks to the Author):

The authors characterized cataplexy induced by sexual excitation in a mouse model of narcolepsy. They found that sexual excitation induced cataplexy more frequently than other triggers, such as chocolate, even during the light period. Most of the cataplexy bouts induced by sexual excitation were preceded by ultrasonic vocalizations (USVs). Based on these observations, the authors proposed that cataplexy in narcoleptic mice is a good behavioral measure to study the brain mechanisms behind positive emotion. The findings are very intriguing. However, because it is known well that positive emotion efficiently induces cataplexy, it is easily imaginable that sexual excitation also induces cataplexy. Therefore, the scientific advances generated by the study are limited.

Response: We agree that our results are in line with expectation. We think important point of our study is that we showed multiple kinds of positive emotion (chocolate and sexual excitation) induced the same behavioral measure (cataplexy) in mice. Such generality has not been reported or expected in other behavioral measure of positive emotion utilized in mice, such as facial expression.

First of all, according to ref 28, the definition of murine cataplexy requires EEG measurements, which were lacking in the current study. Therefore, the behavioral arrests observed may be immature to be called cataplexy.

Response: We did not adopt EEG to avoid possible obstruction of social interaction by the recording cable. Therefore, we call "cataplexy-like behavior" instead of "cataplexy" in the revised manuscript.

It is impressive that a female encounter could induce cataplexy during the daytime, in which chocolate cannot. Is this a qualitative or quantitative difference? Do the authors think that the mechanisms to induce cataplexy are different between these two? Or is the level of excitation and positive emotion much higher with a female than chocolate? If the former is the case, it would be very nice to include some data showing the qualitative difference. The latter possibility suggests that cataplexy is a just measure of the degree of excitement, which may not be a so novel finding.

Response: Of course, sensory input to the brain is different between the two. However, we think output mechanisms from positive emotion to cataplexy would be the same since brain imaging studies in humans suggested that many diverse rewards (food, sex, addictive drugs, music, etc) activate a shared or overlapping brain regions (Neuron 86:646). Even when our assumption (quantitative but not qualitative difference) was correct, we believe our study results are enough worth reporting.

According to Fig. 2, a male encounter could induce cataplexy during the light period, although the frequency was much lower as compared to a female encounter. However, encountering an unknown male may induce negative but not positive emotion. Therefore, I think that this result is surprising considering the inability of chocolate to induce cataplexy during the daytime.

Response: Whether or not encountering an unknown male induce negative emotion is difficult question. However, association between male–male USVs with aggressions has not been reported, and actually apparent fight (bite, or scratch) were not observed in the present study. In our experimental condition, resident AB male only sniffed and chased male intruder. Besides, some reports suggest that the restraint stress reduces male–male USVs (PLOS One 7:e29401, 2012) and that social defeat stress reduces male–female courtship USVs (Physiol Behav 67: 769, 1999). Thus, negative emotions in general inhibit the vocal behavior. Because of this reason, if anything, USVs observed in the present study could be regarded as positive rather than negative emotional expression.

A few human studies suggested that narcolepsy patients show increased responses to positive rewards. In light of this information, did narcoleptic mice emit USV calls more than WT mice did upon a female encounter?

Response: Thank you for your constructive suggestion. Previous reports using WT mice showed 100-300 USVs / min with female encounter (PLOS One 6: e17721, 2011; PLOS One 7: e41133, 2012; Behav Brain Res 256: 677, 2013; iScience 23: 101183, 2020). In the present study, the average number of male–female USVs was 108.3 ± 27.5 / min ($n = 6$) during the first 3 min in the recording period. Thus, ORX-AB mice seemed not more susceptible to positive rewards but within the range of WT mice, at least in the current study paradigm. We calculated the average for 3 min because most studies dealing with courtship USVs adopted relatively short recording time (up to several min) as compared to ours (4 h) and actually USV calls (and incidence of cataplexy-like behavior, see Fig. 1c) decreased as time passed and a mouse probably acclimatized to the encounter. These discussions are included in the revised manuscript.

Reviewer #2 (Remarks to the Author):

This manuscript demonstrates that sexual arousal can trigger cataplexy in orexin ablated mice across the day-night cycle, in contrast to palatable food, which mostly triggers during the night. One challenge in the field is to identify the degree of emotional arousal in animals-- indeed, it is challenging to observe this at all.

This paper finds a clever way to use cataplexy as a kind of readout for emotional arousal. I was particularly impressed by the strong association between the number of vocalizations and subsequent attacks.

Response: Thank you for your positive evaluation on our manuscript.

The paper is straightforwardly written and the figures are easy to understand. The stats are complete and proper, although I

think some of the comparisons could be removed or moved to the figure legend, to make reading a little bit easier. The stats should complement the story and support the conclusion, but in a few places, the stats overwhelm the narrative.

Response: According to the suggestion, some statistics are moved from the main text to the figure legends.

I also appreciate why the authors included the MCH staining; it is fine, but I would hardly expect that anything would be found and I am not sure the manuscript needs it-- the rhythm of cataplexy remained intact to food, suggesting already there was no MCH loss as in the other recent paper.

Response: We used different set of animals between Experiment-1 (chocolate/female/none x day/night) and Experiment-2 (USVs vs. cataplexy). Although we have observed circadian rhythm of cataplexy to food in the Experiment-1 and thought possibility of MCH loss should be very small, we decided to confirm preservation of MCH neurons in the animals used in the Experiment-2.

Reviewer #3 (Remarks to the Author):

The authors are commended for their clear & valuable submission. The experimental designs and overall methodology are described fully & in excellent detail. The figures are well-crafted, showing individual data points where appropriate (e.g., Fig. 1C, etc.). I have only minor comments to modestly improve the overall impact of this generally very good manuscript.

Response: Thank you for your positive evaluation on our manuscript.

1) An experimental design figure will help readers know exactly what the authors did in their studies. Moreover, in this figure, it is advised to include a USV spectrogram example, so that readers unfamiliar with the USV measure can better appreciate what exactly is being measured in the mice.

Response: A schematic explanation was added in Figures 1a and 2a.

2) While the authors acknowledge in the Discussion literature supporting that resident male mice are the likely emitters of USV during recording session, this caveat should be additionally included in the Results text for full transparency. For instance, "ORX-AB mice emitted more USV calls with a female (3688 ± 556 calls)" should be minorly revised to "USVs measured during sessions when ORX-AB mice were exposed to a female were greater ($XXX \pm XXX$) compared to...".

Response: Thank you for your advice. Relevant section was rewritten as suggested.

3) Can a bit more information regarding the acoustic features of recorded USVs be provided? In Methods, it is described that the Avisoft condenser microphone can record frequencies up to 192-kHz. Were the detected USVs predominantly within a frequency range of 40-60 kHz as has been described as relating to "positive affective" USVs in rats? Or were the detected USVs typically much higher in frequency? Additional measures of the recorded USVs would be helpful, minimally including Excellent work!

Response: As the referee 3 pointed out, adult rats emit distress calls (around 22 kHz) and pleasant calls (around 50 kHz). Actually, our USV analysis system "USVSEG" can successfully detects such calls (see, Fig. 3 in Tachibana et al., PLOS One 15: e0228907, 2020). On the other hand, in mice, USVs has not been classified based on mean frequency of main tone. Call pitch of USVs in mice are observed in the frequency range ranging from 60-80 kHz and the mean frequency of calls differ depending on genetic background (PLOS One 6: e17721, 2011; PLOS One 6: e22093, 2011). In revised manuscript, we added the analysis of duration and frequency (see, Fig. 2 and the Result section), and also added discussion as shown below; Observed frequency in both male—female and male—male USVs was comparable (Fig. 2c) and similar to the previously reported values in several genetic strains of mice (PLOS One 6: e17721, 2011; PLOS One 6: e22093, 2011). Meanwhile, average duration of male—female USVs was significantly longer than that of male—male USVs. Long—duration calls of male—female USVs are known to be observed in latter phase of sexual behavior where mounting often occur (PLOS One 11: e0147102, 2016). Therefore, longer duration in the male—female condition here thought to reflect strong sexual excitation.

REVIEWERS' COMMENTS:

Reviewer #1 (Remarks to the Author):

The authors have sufficiently addressed my previous questions and comments.

Reviewer #2 (Remarks to the Author):

The authors have addressed all my concerns.

Reviewer #3 (Remarks to the Author):

The authors have addressed my minor concerns in full. Great work on this interesting & important research.